# Current Knowledge, Research Progress, and Future Prospects of Phyto-Synthesized Nanoparticles Interactions with Food Crops under Induced Drought Stress

Abdul Wahab [1,2,*], Farwa Batool [3], Murad Muhammad [2,4], Wajid Zaman [5,*], Rafid Magid Mikhlef [6] and Muhammad Naeem [7]

1   Shanghai Center for Plant Stress Biology, CAS Center for Excellence in Molecular Plant Sciences, Chinese Academy of Sciences, Shanghai 200032, China
2   University of Chinese Academy of Sciences, Beijing 100049, China; muradbotany1@uop.edu.pk
3   Department of Botany, Lahore College for Women University, Lahore 54000, Punjab, Pakistan; farwabatool176@gmail.com
4   State Key Laboratory of Desert and Oasis Ecology, Xinjiang Institute of Ecology and Geography, Chinese Academy of Sciences, Urumqi 830011, China
5   Department of Life Sciences, Yeungnam University, Gyeongsan 38541, Republic of Korea
6   Biotechnology Department, University of Samarra, Samarra 34010, Iraq; rafid.magid@uosamarra.edu.iq
7   Department of Plant Science, School of Agriculture and Biology, Shanghai Jiao Tong University, 800 Dongchuan Road, Shanghai 200240, China; naeembbt@gmail.com
*   Correspondence: wahabcrop_science@mails.ucas.ac.cn (A.W.); wajidzaman@yu.ac.kr (W.Z.)

**Abstract:** Drought stress threatens global food security and requires creative agricultural solutions. Recently, phyto-synthesized nanoparticles NPs have garnered attention as a way to reduce food crop drought. This extensive research examines how phyto-synthesized NPs improve crop growth and biochemistry in drought-stressed situations. The review begins with an introduction highlighting the urgency of addressing the agricultural challenges posed by drought. It also highlights the significance of nanoparticles synthesized from photosynthesis in this context. Its purpose is to underscore the importance of sustainable farming practices. This approach is contrasted with conventional methods, elucidating the ecological and economic advantages of phyto-synthesized NPs. This review discusses phyto-synthesized nanoparticles, including titanium dioxide, iron oxide, gold, silver, and copper. In addition, we review their ability to enhance crop growth and stress resistance. The primary focus is to elucidate the effects of phyto-synthesized NPs on plant development under drought stress. Noteworthy outcomes encompass improvements in seed germination, seedling growth, water absorption, photosynthesis, chlorophyll content, the activation of antioxidant defense mechanisms, and the modulation of hormonal responses. These results underscore the potential of phyto-synthesized NPs as agents for enhancing growth and mitigating stress. The review assesses the risks and challenges of using phyto-synthesized NPs in agriculture. Considerations include non-target organisms, soil, and environmental impacts. Further research is needed to determine the long-term effects, dangers, and benefits of phyto-synthesized NPs. Nanoparticles offer a targeted and sustainable approach for improving plant drought tolerance, outpacing traditional methods in ethics and ecological balance. Their mechanisms range from nutrient delivery to molecular regulation. However, the long-term environmental impact remains understudied. This review is critical for identifying research gaps and advancing sustainable agricultural practices amid global water scarcity.

**Keywords:** phyto-synthesized NPs; agriculture; drought stress; plant growth; biochemical attributes; environmental impact

## 1. Introduction

### 1.1. Drought Stress and Its Impact on Plant Growth and Development

Drought stress occurs in plants with an internal water deficit due to an inadequate water supply. The condition manifests when plants are without sufficient water for an extended period. Plants' physiology, morphology, and productivity can all be drastically altered by drought stress. One significant effect of drought stress is reducing crop yields [1,2]. Plants cannot carry out their critical physiological functions when deprived of water. The plant's photosynthesis, nutritional uptake, and hormonal balance may all suffer, causing stunted growth, withering, and even death [3,4]. Drought stress also interferes with many plant processes. Many biological activities, such as cellular respiration, which produces energy, require water. These mechanisms, metabolic activity, and plant health, in general, suffer when water is scarce [5–7]. Stomatal apertures, which regulate the exchange of gases and water vapor between the plant and the environment, are similarly affected by drought stress. Increased transpiration can worsen water shortages caused by this interruption [8,9].

Drought stress causes plants to adapt physiologically by altering gene expression, accumulating Osmo protectants (such as proline and carbohydrates), and activating stress-responsive signaling cascades. These systems can aid plants in dealing with drought stress, but they are usually insufficient to prevent serious crop production decreases [10,11]. Understanding the effects of drought stress on plants and the processes disrupted under these conditions is essential for developing solutions to alleviate the detrimental repercussions of drought stress on agriculture and the environment. Understanding the complex mechanisms involved in the drought stress response might help researchers and agricultural practitioners increase crop resilience, enhance water management practices, and guarantee food security in the face of more unpredictable climatic conditions [12–15].

### 1.2. Why Phyto-Synthesized NPs Are Promising

Phyto-synthesized NPs are a diverse group of nanoparticles, economically crucial for the production of crops and resilience against drought conditions. Phyto-synthesized NPs have shown many unique properties by increasing efficiency and surface-area-to-volume ratios. This leads to beneficial nutrient uptake and effects on growth, breaches the biological blocks, and links with plant organisms at the molecular level [16,17]. Previous studies showed that specific photo-synthesized NPs could improve the photosynthetic ability of plants, letting them cope with water-stress conditions and capture and utilize light more efficiently. Moreover, this improvement in photosynthesis makes plants capable of surviving during drought [18,19]. It accelerates their growth and productivity, thus leading to abundant agricultural yields. Silver nanoparticles are potent candidates for applying phyto-synthesized NPs in agriculture [18,20–23]. Silver nanoparticles have been shown to enhance the water retention capacity of soils, reducing water loss through evaporation and increasing water availability for plant uptake during dry periods [24–26].

Moreover, these nanoparticles can stimulate the expression of stress-related genes in plants, leading to improved drought tolerance and better adaptation to challenging environmental conditions [27]. Silver nanoparticles are the most critical candidate for phyto-synthesized NPs in agriculture [28–30], according to previous research, which explains that this particle showed the best results by increasing the water retention capability of soils, decreasing water loss through evaporation, and enhancing the availability of water uptake during drought conditions [31,32]. Functionally, these particles can regulate the expression of stress-related genes in plants, helping in drought tolerance and adaptations against challenging environmental conditions [33,34].

In conditions like less nutrient availability and uptake to plants in arid soils, encapsulated nutrients in nanoparticles are used to ensure their targeted delivery in the roots of plants. Regardless of the encouraging benefits of nanoparticles in the agricultural field, it is hard to consider their potential risks and effects on the environment [35,36]. Nanoparticles can interact with various microorganisms and beneficial insects. Therefore, their long-lasting effects should be deeply investigated. When incorporated into agricultural



settings, these nanoparticles should not harm the ecosystem, biodiversity, and human well-being [35–37]. Regulatory authorities and researchers must oversee the risk assessment to calculate the potential environmental influence of nanoparticles in the agricultural sector [38–40]. According to prior studies, the changes in the soil, their nutrient uptake and accumulation in plant cells, and their effects on other non-targeted organisms are important [41–43]. Phyto-synthesized NPs counter abiotic conditions like drought and less crop production. Nevertheless, systematic research is necessary to observe their mode of action, potential threats, and related risks to check their potential and ensure their safe use fully [44,45]. The outcomes of prior research indicate that these particular aspects and particles possess the potential to serve as integral instruments for enhancing worldwide food security during the epoch of climate fluctuations and the burgeoning of the human populace [46–48].

### 1.3. Review Scope and Approach

This review focuses on the characterization and synthesis of various phyto-synthesized NPs, their responses to drought conditions, and their effects on plant growth. The study also explores molecular mechanisms and environmental concerns. It identifies research gaps for future investigation in utilizing various phyto-synthesized NPs to tackle agricultural challenges amidst global climate change. For this study, we thoroughly reviewed the most recent literature on the characteristics of nanoparticles. Our investigation involves a thorough comparison of these results with the field's present and projected futures. Reputable academic databases like Google Scholar, PubMed, Elsevier, and the NCBI official repository were used to research this time period, which runs from 2017 through 2023. The discussion encompasses diverse nanoparticles, their synthesis methods, growth impacts, water relations, antioxidant properties, and interactions with the photosynthesis system. Additionally, potential challenges and risks in agricultural applications are acknowledged. This neutral and comprehensive review contributes valuable insights for researchers and practitioners in the field.

## 2. Phyto-Synthesis of Nanoparticles

### 2.1. Green Synthesis Approach

In recent years, the green synthesis approach has emerged as an eco-friendly and promising phyto-synthesized NP method. The inventive technique includes using plant-derived elements by reducing and stabilizing the agents to produce nanoparticles of different materials, like metal and metal oxide [49–51]. This synthesis deals with numerous benefits, such as cost-effectiveness and sustainability. It decreases the potential threats linked to traditional chemical methods. A significant example of the green synthesis approach is the application of plant extract and bioactive compounds such as alkaloids, terpenoids, polyphenols, and flavonoids as reducing agents in the synthetic process, as shown in Table 1. These biomolecules of plant extracts have intrinsic properties, making them capable of reducing metal ions and helping produce nanoparticles [10,52–55].

**Table 1.** Comparative Analysis of the Green Synthesis of Nanoparticles: Methodological Approaches and Implications for Sustainability and Practical Applications.

| Category | Description | References |
| --- | --- | --- |
| Methodology | Utilization of biological entities such as plants or plant extracts in the formation of nanoparticles | [56–58] |
| Environmental Impact | Lower toxic byproduct generation, environmentally friendly process | [52,59] |
| Cost-Effectiveness | Cost-effective compared to chemical synthesis | [53,54] |
| Application Areas | Used in medicine, electronics, catalysis, environmental remediation | [55,60] |
| Scalability | Adaptable to various applications | [26] |
| Time Efficiency | Generally, a faster process | [61] |
| Safety Considerations | Safer synthesis process, minimizing risks | [62] |

Moreover, the extract acts as a stabilizing agent, inhibiting agglomeration and ensuring the stability of nanoparticles by increasing their capability and applicability in different fields. Furthermore, using plant-based materials to synthesize nanoparticles is a sustainable alternative to conventional methods using toxic chemicals and high energy. This approach of plant-derived compounds is readily available and biodegradable, contributing to the friendly environment [63–65]. These green synthesized nanoparticles proved very applicable in drug delivery, catalysis, environmental remediation, and agriculture. According to previous studies, silver nanoparticle synthesis from leaf extracts of certain plants showed excellent antimicrobial properties, demonstrating that these plants have strong antibacterial properties [63–65]. Similarly, the iron oxide nanoparticles synthesized by the green synthesis method showed the properties of removing pollutants from the wastewater, allowing a green solution for water treatment. Hence, the green synthesis method is a more sustainable and promising avenue for nanomaterial production [66–68].

## 2.2. Advantages of Phyto-Synthesis over Other Methods

Phyto-synthesis has developed as an encouraging and sustainable alternative to conventional synthetic methods, putting away beneficial interests among industries and scientists. The significant benefit of Phyto-synthesis is the reduction in energy consumption compared to other physical and chemical conventional methods [69–71]. Practice plant extract is a stabilizing and reducing agent, minimizing the need for high-temperature and energy-intensive processes. This approach is cost-saving and helps with achieving a greener community by reducing nanoparticles' overall carbon footprint production [72,73]. The critical advantage of phyto-synthesized NPs is biocompatibility, as plant extracts are used for synthesis. The obtained nanoparticles are inherently more compatible with biological systems [74,75]. It opens up various applications in medicines like drug delivery systems, where the human body more freely consumes the nanoparticles by reducing potential adverse reactions. One major factor is the production of toxic-free end products in photosynthesis, as described in Table 2 [76–78].

**Table 2.** Advantages of phyto-synthesized NPs over other methods.

| Advantage | Phyto-Synthesis | Other Methods | Reference |
| --- | --- | --- | --- |
| Environmental Friendliness | Low toxic byproduct generation | Hazardous waste | [76] |
| Cost | Cost-effective | Expensive reagents | [77] |
| Energy Efficiency | Less energy consumption | More energy consumption | [70] |
| Scalability and Versatility | Adaptable | Limited scalability | [79] |
| Use of Renewable Resources | Renewable plant materials | Non-renewable materials | [80] |
| Time Efficiency | Faster process | Time-consuming | [81] |
| Safety Considerations | Safer due to non-toxic materials | Safety risks | [82] |
| Quality Control | Fine control | Quality variations | [83] |

Further, most processes end with hazardous chemicals, which may lead to high risks to human health and cause environmental pollution. In contrast to phyto-synthesized NPs and chemical extracts, prior ones are safe for researchers and workers and eco-friendly for the environment [79]. The previous study utilized green phyto-synthesis to produce silver nanoparticles using Aloe vera leaf extract. The scientists combine the plant extract with a sliver salt solution and allow the mixture to undergo a heating process. The resulting products maintain excellent biocompatibility and stability, making them suitable nanoparticles for active biomedical applications, such as wound-healing agents and antibacterial properties [84,85]. The green synthesis process also offers better control over the size and shape of nanoparticles, a critical aspect for modifying their characteristics for specific use and function. Phyto-synthesis has an excess of advantages over other conventional methods. It is environmentally friendly, biocompatible, has no toxic chemicals, and decreases energy consumption costs, providing a promising path for sustainable production of nanoparticles. Soon, due to deep research to understand plant-based synthesis methods and explore new

plant resources, we can advance with more innovative advancements and applications in this field [86,87].

## 3. Phyto-Synthesized-NPs and Their Applications in Agriculture

Various phyto-synthesized NPs have recently gained attention in the agricultural field. The use of these nanoparticles in agriculture proved to be more efficient and convenient for the environment and provide good crop health compared to other conventional synthetic nanoparticles. These play a vital role in avoiding toxic environmental components, keeping the ecosystem preserved, and safeguarding the health of consumers and farmers [10,88]. One significant advantage of various phyto-synthesized NPs is their safety usage. In contrast to local synthetic nanoparticles that indeed have unknown prolonged effects, green synthetic nanoparticles have fewer adverse issues for human health and the environment [89,90]. The alternative significant factor is cost-saving, which drives their practice in the agricultural field. Although the synthetic process relies on plant extracts, the production cost is relatively low. In addition, as shown in Table 3, these nanoparticles become readily available to many farmers, including in developing countries, to improve agronomic practices globally [60,63].

**Table 3.** Overview of the Influence of a Multitude of Nanoparticles on Enhancing Drought Tolerance in Plants.

| Plant Species | Nanoparticle | Observed Effect | References |
|---|---|---|---|
| Wheat | Iron-NPs | Enhanced photosynthesis and reduced oxidative stress | [91–93] |
| Wheat | Silica-NPs | Enhanced plant growth and development | [94] |
| Rapeseed | Maghemite-NPs | Increased growth and reduced water stress | [95,96] |
| Wheat | Selenium-NPs | Improved plant growth and development | [84,94,97] |
| Wheat | Iron-NPs | Increased plant growth and reduced oxidative stress | [97,98] |
| Corn | $TiO_2$-NPs | Increased water use efficiency | [99] |
| Sunflower | Silver and $TiO_2$-NPs | Improved root and shoot length | [100–102] |
| Soybean | Copper oxide-NPs | Mitigated drought effects via increased antioxidant activity | [103–105] |
| Rice | Alumina-NPs | Promoted nutrient uptake and improved drought resilience | [106,107] |
| Cotton | Chitosan-NPs | Enhanced leaf water content and reduced transpiration rate | [108,109] |
| Barley | Zirconia-NPs | Increased root length and biomass | [110,111] |
| Tomato | Cerium oxide-NPs | Improved stomatal conductance and water use efficiency | [112,113] |
| Maize | Gold-NPs | Enhanced photosynthetic efficiency and reduced oxidative stress | [114,115] |
| Chickpea | Silica-NPs | Augmented nutrient uptake and drought tolerance | [116,117] |

Moreover, the versatile nature of phyto-synthesized NPs opens up various applications in farming. The most important feature is their antimicrobial activity, which controls diseases linked to weeds, insects, and pests in crops [66]. With various phyto-synthesized NPs, farmers can decrease the resilience of harmful pesticides by promoting safe cultivation practices and reducing the issues linked with chemical exposure [118,119]. When harnessed into the soil, these will enhance nutrient availability, resulting in better plant growth and higher yields. The presence of nanoparticles in the soil can help against soil erosion, increasing sustainable land management practices [120,121]. By understanding their potential threats, recent research has shown the successful preparation of silver nanoparticles synthesized from the *Azadirachta indica*, natively known as the Neem tree. The plant extract of the Neem tree controls fungal pathogens in crops [122,123]. Research showed excellent activity against fungal pathogens and enhanced stability, constructing them as a valuable and efficient way of disease management in agriculture. It has opened new horizons in the agronomy field for sustainable development goals. Safe usage, environmental friendliness, affordability, and various applications make them a reliable asset in advanced cultivation practices [124–126]. However, researchers contribute to exploring new potential. We hope for future advancements and innovative usage of nanoparticles to transform agricultural ways and subsidize greener and cleaner crop production. Table 4 presents comprehensive

information and explains the structure of nanoparticle-reduced drought stress damage and adverse effects in plants [127,128].

**Table 4.** Types of Nanoparticles and Their Mechanisms of Action for Mitigating Drought Stress in Specific Food Crops.

| Nanoparticle | Mechanism of Action | Applications in Drought Stress Mitigation | Specific Food Crops Tested | References |
|---|---|---|---|---|
| Silver Nanoparticles | Enhance water uptake, modulate stress-responsive genes | Improved drought tolerance, water use efficiency | *Oryza sativa* L. | [129,130] |
| Gold Nanoparticles | Improve root architecture, influence osmotic adjustment | Enhanced root growth under drought conditions | *Triticum aestivum* L. | [131,132] |
| Zinc Oxide Nanoparticles | Increase antioxidant enzymes, regulate stress-related hormones | Protection against oxidative damage under drought stress | *Zea mays* L. | [133,134] |
| Iron Oxide Nanoparticles | Improve water utilization efficiency, enhance chlorophyll content | Enhanced growth and yield under water-limited conditions | *Glycine max* L. *Merr.* | [135,136] |
| Copper Oxide Nanoparticles | Regulate stomatal conductance, improve nutrient uptake | Increased resistance to water stress, improved nutrient efficiency | *Sorghum bicolor* L. *Moench* | [136,137] |
| Titanium Dioxide Nanoparticles | Increase water retention, enhance photosynthetic efficiency | Improved growth and reduced water loss under drought conditions | *Hordeum vulgare* L. | [99,138,139] |
| Silica Nanoparticles | Enhance cell wall rigidity, increase water holding capacity | Reduced evapotranspiration, improved drought tolerance | *Solanum lycopersicum* L. | [33,140] |
| Cerium Oxide Nanoparticles | Act as antioxidants, regulate water and nutrient transport | Enhance drought resistance by reducing oxidative stress | *Cucumis sativus* L. | [103,141,142] |
| Magnesium Oxide Nanoparticles | Improve nutrient absorption, enhance water use efficiency | Increased drought tolerance, improved nutrient availability | *Lactuca sativa* L. | [143,144] |
| Calcium Oxide Nanoparticles | Regulate cell membrane stability, increase root water uptake | Enhanced root development, improved water retention | *Spinacia oleracea* L. | [145,146] |
| Aluminum Oxide Nanoparticles | Modulate stress-responsive enzymes, increase water retention | Improved growth and stress tolerance under drought conditions | *Helianthus annuus* L. | [147–150] |
| Selenium Nanoparticles | Act as antioxidants, enhance stress tolerance mechanisms | Improved growth, yield, and water use efficiency under drought conditions | *Brassica napus* L. | [151,152] |

Several methods can synthesize nanoparticles, including chemical vapor deposition and sol–gel synthesis, to achieve specific particle properties. Post-synthesis treatments may further refine these characteristics after the completed synthesis [58]. X-ray diffraction and scanning electron microscopy are commonly used to evaluate nanoparticles' physical and chemical properties. The suitability of nanoparticles for application depends on the synthesis and the characterization of the particles [55]. We outline some examples of nanoparticles that help plants cope with drought stress conditions and alleviate their adverse effects in the following sections.

### 3.1. Silver Nanoparticles (Ag-NPs)

Silver nanoparticles showed relatively good antifungal and antimicrobial properties, assisting them to control different diseases of plants and enhance their production rate. One convenient method for synthesizing Ag-NPs involves using various plant extracts, such as *Aloe Vera*, *Azadirachta indica*, and *Ocimum sanctum* [153–155]. Ag-NPs proved very useful against multiple stresses, but drought or water stress is the most important. Recent studies demonstrated that silver nanoparticles enhanced the *Fusarium oxysporum* growth rate of a root, leaf area, and shoot [156–158]. The previous studies also showed that Ag-NPs synthesized with *Justica adhatoda* leaf extract could improve chickpeas' growth rate and yield by improving photosynthesis, antioxidant activity, and nutrient uptake [159]. Several articles have recently been explored for testing Ag-NP's response to kill off dangerous bacteria and fungi and inhibit crop growth. As a result, these nanoparticles proved to be an ideal candidate in the agricultural field, where pathogens can frequently affect the crop's growth, yield, and quality [160–163]. Ag-NPs proved the best alternative to various synthetic and chemical pesticides. Specifically, the leaf extract of *Moringa oleifera* can fight against bacterial blight disease in pomegranate trees, usually caused by *Xanthomonas axonopodis* bacteria [164,165].

Earlier research concluded that their findings proved safe alternatives to chemical-based sprays to inhibit insecticide growth and create an eco-friendly environment. Another extract from *Justicia adhatoda* leaf showed the properties of improving growth rate in chickpeas [166–168]. Applying Ag-NPs on chickpeas revealed the positive aspects of increased photosynthesis, antioxidant activities, and nutrient absorption. Moreover, when silver nanoparticles were extracted from the leaf of *Carum copticum*, plant development increased by increasing sunlight absorption and nutrient availability and reducing the damage caused by oxidation [169–172]. Ag-NPs are very helpful in the production of the wheat crop by enhancing wheat grains' quality and safety from different fungal attacks. Suggestions might be the effectiveness and protection of these NPs in the most popular staple food crop (wheat) globally [172–175]. Ag-NPs can be used as safe fertilizers in agricultural sectors instead of chemical fertilizers. These NPs can cause several potential impacts on the environment and regulatory issues, so it is necessary to understand their mode of action in promoting plant growth and disease control actions [85,176–178].

### 3.2. Gold Nanoparticles (Au-NPs)

In drought conditions of plants, the use of gold nanoparticles helps the plants to grow and also facilitates the process of photosynthesis. Moreover, Au-NPs have been shown to have incredible features in protecting plants from their oxidative reactions caused by drought conditions through anti-oxidative activity [179–181]. To synthesize Au-NPs, some plant extracts such as green tea, grape seeds, and lemon peel are more efficient to a large extent. During dry seasons, plants can improve performance and physiology when they contact gold particles—specifically, growth rate and yield increase when green tea extract synthesizes Au-NPs [182–184]. According to Wahab et al., 2023, Au-NPs help plants to take up nutrients efficiently from soil and enhance their photosynthetic pigments that provide the best food resources for plants [185]. According to El-Saadony et al., 2022, using grape seeds for gold nanoparticles showed increased nutrient absorption, photosynthesis, antioxidant process, growth, and best yield quality in wheat crops [186]. During drought stress, when plants can easily be affected by microbes, the gold nanoparticles proved to help fight against microbes due to antimicrobial activity by inhibiting infection rates caused by pathogens. The fundamental features of Au-NPs are to combat disease and increase the immunity system in plants. The growth and development of plants in contact with Au-NPs enhance the ability to penetrate plant tissues, allowing [187,188].

Moreover, Au-NPs not only help deliver micronutrients but also improve the germination of seeds and the growth rate of tissues. Au-NPs may also increase plant development by controlling gene expression and their activating pathways [189]. Above and beyond influencing the expression of a gene, Au-NPs might also cooperate with plant membranes

and enhance the ability of drought tolerance. However, the exact process of their activity is still unknown. For scavenging oxygen-reactive species, gold nanoparticles fight oxidative damage triggered during drought conditions [190–192]. There are some safety measures when implanting Au-NPs in plants during drought conditions, such as checking their potential environmental impacts. Scientists are working on Au-NPs for their destiny and build-up in the atmosphere; their impressions on soil and plant systems and their possible threats to non-targeted species are also part of continuing research [193,194]. Au-NPs extracted from plant bioactive compounds have emerged as a potential gizmo for increasing vegetative resilience against dry weather. Their capability to enhance nutrient uptake, anti-oxidative properties, photosynthetic rate, and disease resistance make them cherished for cultivation applications. Furthermore, there is a need for new and entirely understandable research on their mode of action, controlled application procedures, and eco-friendly nature for general approval in drought conditions [195–197].

### 3.3. Iron Oxide Nanoparticles (Fe$_3$O$_4$-NPs)

Due to their vast characteristics, Fe$_3$O$_4$-NPs are increasingly efficient in agronomy for drought management. These nanoparticles are highly effective and extremely useful for the safety of soil and plants [198]. Fe$_3$O$_4$ nanoparticles are very peculiar for absorbing and adsorbing. They can hold a large amount of water, therefore proving helpful in drought-stressed regions [199]. On the other hand, Fe$_3$O$_4$ nanoparticles provide a protective covering for plants and boost crop quality and yield in drought conditions [200,201]. Most studies have shown that Fe$_3$O$_4$-NPs can hold water in the soil, increase the photosynthetic rate, and reduce water evaporation. This enables plants to grow under drastic environmental conditions. Also, Fe$_3$O$_4$ nanoparticles are environmentally friendly and can manage high drought stresses. One major factor is to sustain a firm basis with non-toxic and biodegradable compounds [29,202,203]. Fe$_3$O$_4$-NPs are also cost-saving and more appealing to farmers for better crop production and development. Concluding the features of Fe$_3$O$_4$-NPs, they may be a promising tool for decreasing drought stress in cultivation systems by providing a safety coating to soil and plants from the atmospheric barriers. These are eco-friendly and money-saving, making them an ideal choice for cultivars in their farming sectors [204–206].

### 3.4. Copper Nanoparticles (Cu-NPs)

In recent years, Cu-NPs have gained attention for their potential use for stress management in the agricultural sector. A particularly significant way is using plants' bioactive compounds to synthesize Cu-NPs. Thus, their eco-friendly nature provides various beneficial aptitudes to plant crops [80,207]. Previous studies showed that Cu-NPs synthesized from plant materials positively impact the development and growth of plants. Similarly, these proved to be a protector against challenging environmental conditions like dryness in the atmosphere. Case in point, when tomato plants were treated with copper nanoparticles synthesized from the basil leaf extract, Tripathi et al. (2022) unveiled properties like photosynthesis and antioxidant activity under drought conditions [4,208,209]. The presence of nanoparticles acts as a protective shield against oxidation reaction damages produced by unfavorable conditions. Therefore, it helps plants balance optimal metabolic functions and photosynthetic rates during water scarcity [210–212].

Moreover, Cu-NPs were found to be valuable in modulating the response of the genes associated with the stress response in plants. It also contributes to their efficiency in fighting against environmental alterations. The gene regulation activity assists plants in activating their stress tolerance mechanism, including the production of proteins and Osmo protectants that aid in cellular protection and water retention [213,214]. For instance, crops treated with Cu-NPs will more likely reveal increased drought resistance by enhancing overall productivity. The eco-friendly and money-saving feature of this green-synthesis method revealed the positive impacts on the development of the plant in stress response conditions and photosynthesis [215,216]. It attracted the attention of researchers and

potential enactment in agricultural practices. Cu-NPs have significant attributes in different fields due to their distinctive physiochemical and potential applications [59,217].

Consequently, the typical range of a nanoparticle varies from 1 to 1000 nanometers. It retains exceptional catalytic activity, antimicrobial properties, and electrical and thermal conductivity. There are many ways to synthesize Cu-NPs, such as green synthesis, chemical reduction, and Sono chemical processes posing control over the surface's shape, size, and functionality to develop their characteristics for specific applications [218,219].

This review article extensively covers diverse applications of copper nanoparticles (Cu-NPs) across various fields. In microchip technology, Cu-NPs find utility as conductive inks for printed electronics and as additives to enhance the performance of electric devices [220,221]. Their inherent ability to combat microbes in the medical sphere positions them as ideal candidates for biomedical and wound-dressing applications. Leveraging the catalytic potential of Cu-NPs in environmental remediation unveils their proficiency in breaking down pollutants and facilitating the safe extraction of heavy metals present in contaminated water sources. Owing to their multifunctional attributes, many possibilities emerge for integrating these nanoparticles into various technologies and industries, catalyzing further research and experimentation [200,203,222].

### 3.5. Zinc Oxide Nanoparticles (ZnO-NPs)

In previous research endeavors, the significant merits of ZnO-NPs extracted from plant materials have garnered attention due to their innovative role in facilitating plant development and their capacity to endure challenging environmental conditions [223–225]. Within global warming, heightened concern surrounds the impact of drought stress on the agricultural sector, manifesting as reduced productivity and posing challenges in adequately nourishing the global population [226,227]. For example, an inclusive review by Gupta et al. in 2022 underscored that applying ZnO-NPs to drought-exposed plants elicits heightened plant growth performance rates, concurrently bestowing a spectrum of notable functions. Notably, these nanoparticles facilitate augmentation in chlorophyll content, thereby fostering an upswing in photosynthetic activity [228,229]. Consequently, this augmentation translates to an enhanced energy storage and conservation capacity, even with limited water availability. Applying Zinc oxide nanoparticles (ZnO-NPs) has been demonstrated to effectively regulate the activity of antioxidant enzymes such as catalase and superoxide dismutase [230–232]. These enzymes are crucial in counteracting oxidative damage caused by accumulating reactive oxygen species (ROS), particularly under drought-induced conditions. ZnO-NPs function as protective barriers for essential cellular components, enabling plants to withstand the adverse impacts of drought by mitigating oxidative reactions [103,233,234]. Remarkably, ZnO-NPs offer an additional advantage to plants by enhancing their resilience to arid conditions at the molecular level. Despite the direct interaction of nanoparticles with plant cells, they can modulate the expression of stress-responsive genes. This modulation subsequently triggers a cascade of defense mechanisms and adaptable responses, ultimately enhancing the plants' capacity to tolerate drought [202,210,235]. Notably, the impact of ZnO-NPs on crucial genes associated with the abscisic acid (ABA) signaling pathway, pivotal in plant water stress responses, has been investigated. ZnO-NPs facilitate ABA-dependent pathways, thereby aiding in the closure of stomata. Consequently, this closure reduces water loss through transpiration, conserving water during drought stress [107,236].

Furthermore, ZnO-NPs influence the gene that encodes the enzyme involved in osmotic adjustment and proline biosynthesis. Alterations induced by ZnO-NP applications lead to elevated proline levels, enhancing the plant's water absorption capability. This augmentation positively affects the plant's water status and adaptability to drought conditions. A critical perspective to consider is that utilizing ZnO-NPs derived from plants presents an environmentally friendly and ecologically sustainable strategy to mitigate the adverse effects of drought stress on agricultural practices [237–239]. The multifaceted advantages of ZnO-NPs encompass the enhancement of chlorophyll function and antioxidant enzyme

activity and the stimulation of stress-responsive genes, thereby fostering plant development and augmenting drought tolerance [240].

*3.6. Titanium Dioxide Nanoparticles (TiO$_2$-NPs)*

TiO$_2$-NPs have demonstrated their potential as beneficial and promising tools in the agricultural domain, primarily due to their positive effects on plant development, growth, and stress tolerance capabilities [241,242]. Like other nanoparticles, TiO$_2$-NPs can be synthesized from plant extracts, revealing noteworthy growth rates, improved photosynthetic plant efficiency, and enhanced antioxidant activities. A specific study involving tomato plants highlighted that applying TiO$_2$-NPs resulted in heightened chlorophyll content, increased biomass, and elevated food production rates under drought conditions with limited water availability [243–245]. Furthermore, TiO$_2$-NPs have effectively alleviated oxidative stress conditions caused by environmental adversities. These nanoparticles have shown significant scavenging abilities against reactive oxygen species (ROS), which tend to accumulate in plant tissues during drought conditions, detrimentally impacting plant health, growth, and cellular integrity [246,247]. Salam et al. (2022) reported that treating rice plants with TiO$_2$-NPs reduced ROS levels, enhancing stress tolerance and augmenting crop productivity, particularly in high salinity conditions [248,249]. The multifunctionality of TiO$_2$-NPs within plants positions them as valuable assets for promoting sustainable agricultural growth. These nanoparticles can counteract the adverse effects of abiotic stressors on plant physiology, fostering overall plant health and development [243]. This aspect assumes greater significance in addressing global food security concerns by bolstering the resilience of the agricultural sector. Over time, researchers have gained a more comprehensive understanding of the underlying mechanisms and potential environmental implications of utilizing TiO$_2$-NPs in agronomy [244,250,251]. TiO$_2$-NPs derived from natural plant materials offer promising attributes for enhancing plant growth, fortifying antioxidant defense mechanisms, and optimizing photosynthetic efficiency under stressful conditions.

*3.7. Nanoparticle–Plant Interactions: Mechanisms of Uptake, Translocation, and Implications for Agricultural Applications*

It is important to note that the scholarly discussion about how nanoparticles enter and move through different crops is based on specific plant species and nanoparticle types, which have already been discussed in a thorough review. The root system is a significant way that nanoparticles can be taken up by more than one crop. When a plant's roots take Silver Nanoparticles in, they help it take in more water and change how genes that respond to stress work. This happens in plants like Sorghum (*Sorghum bicolor* L.) and Corn (*Zea mays* L.). Nanoparticles move through plants in two main ways: Apoplastic and symplastic. Wheat (*Triticum aestivum* L.) roots can grow better during drought because of these routes. The leaf can also take in nanoparticles like Titanium Dioxide or Silica through foliar uptake. Potato (*Solanum tuberosum* L.) plants can keep more water and make more food by spraying Titanium Dioxide Nanoparticles on them as a fog.

The xylem and phloem of plants help nanoparticles move over long distances. Zinc oxide nanoparticles that travel through the xylem to other parts of the plant help control stress hormones and enzymes that fight free radicals. Using nanoparticles like Iron Oxide and Copper Oxide has been shown to improve how well spinach (*Spinacia oleracea*) uses water and how much chlorophyll it has and to make lettuce (*Lactuca sativa*) more resistant to water stress. The physicochemical features of the nanoparticles and the environment change how these complicated dynamics work. Silica is found in rice (*Oryza sativa*). The size and charge on the surface of nanoparticles affect how they are taken in and distributed, making cell walls stiffer and helping them keep water.

Nanoparticle uptake and movement inside the plant depend on the type of nanoparticle and crop type. Nanoparticles like silver, gold, and zinc oxide have been shown to affect Sorghum, Corn, and Wheat differently, depending on whether they are taken in through the roots or the leaves and whether they move through apoplastic or symplastic paths.

Scientists from different fields must work together to fully understand how nanoparticles and plants interact. This is because physical and environmental factors also affect this complex interaction. Understanding this is important to make nanoparticles useful in sustainable agriculture and improve risk assessment procedures.

These qualities make them valuable instruments in the field of agronomy. Figure 1 shows that despite these promising results, further research using well-planned experiments is essential to guarantee the safety and efficacy of introducing NPs into agricultural methods [252–254].

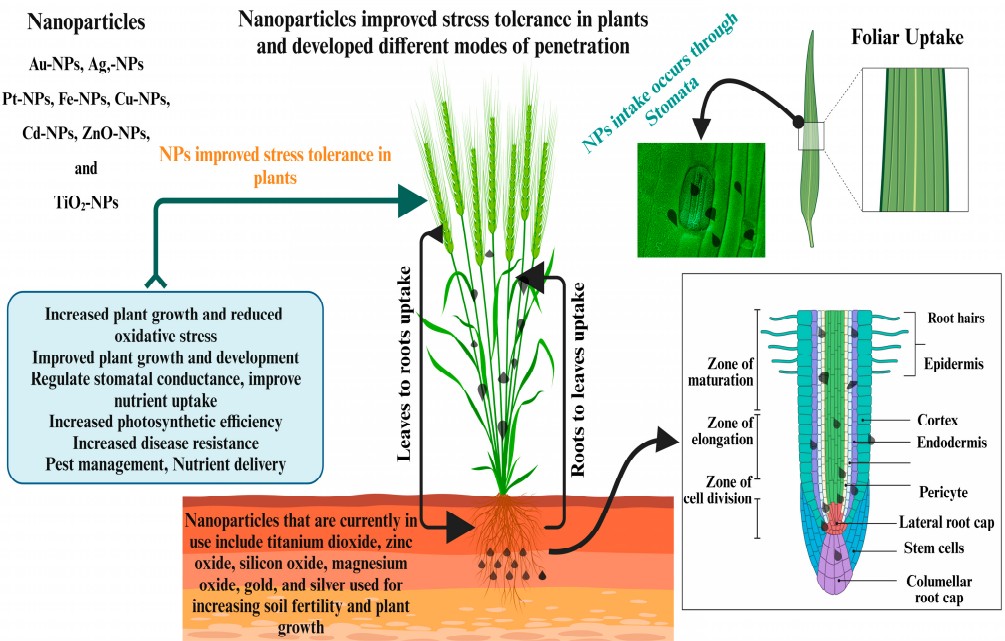

**Figure 1.** Nanoparticles improved stress tolerance in plants and developed different modes of penetration.

## 4. Mechanisms of Nanoparticle-Mediated Drought Stress Alleviation

These methods guarantee reproducible, practical, and optimized results. The essential laboratory method of in vitro culture requires a precise sequence of steps [103,255,256]. One must avoid contamination by carefully selecting and sterilizing explants based on plant tissue type. Each plant species has different nutritional needs, so the culture medium includes nutrients, vitamins, and plant growth regulators [257,258]. The explants are first placed on the culture medium, and subculturing ensures growth and differentiation. Shoot, root, and somatic embryo regeneration requires precise hormonal balances. Finally, acclimatization helps regenerated plantlets adjust to life outside the lab. Modern cryopreservation uses multiple methods to preserve plant genetic resources. Most plans use vitrification and controlled freezing. Ice crystals can damage cells, but controlled freezing can reduce them, as shown in Table 5 [210,259,260].

**Table 5.** Multifaceted Mechanisms of Nanoparticle-Induced Drought Resilience in Agriculture".

| Mechanism | Description | Example Nanoparticles | Potential Applications | References |
|---|---|---|---|---|
| Water Retention | Nanoparticles can enhance soil's water-holding capacity, thus reducing water loss during periods of drought. | Hydrogel, Clay | Crop growth and development | [261,262] |
| Nutrient Delivery | Nanoparticles can encapsulate and deliver essential nutrients to plants, ensuring optimal growth even in water-scarce conditions. | Chitosan, Silica | Crop Growth, Soil Health | [263,264] |
| Stress Signal Modulation | Nanoparticles can be engineered to interact with plant signaling pathways, helping plants manage drought stress better. | Gold, Silver | Plant Stress Management | [265,266] |
| Controlled Irrigation | Nanoparticles can be used in innovative irrigation systems to respond to soil moisture levels and provide controlled and efficient water delivery. | Polymer-based | Precision Agriculture, Water Saving | [267,268] |
| Enhanced Root Growth | Specific nanoparticles can promote root growth, allowing plants to access deeper water sources and better withstand drought conditions. | Carbon Nanotubes | Agriculture, Reforestation | [266,269,270] |
| Photoprotection | Nanoparticles can shield plants from excessive sunlight, which often accompanies drought conditions, thereby reducing stress and damage to the plant. | Titanium Dioxide | Sun Protection for Plants | [271,272] |
| Stimulating Microorganisms | Nanoparticles can foster the growth of beneficial soil microorganisms, enhancing soil structure and increasing water retention capacity. | Zinc Oxide | Soil Health, Microbial Enhancement | [273,274] |

Vitrification prevents ice formation by rapidly cooling water to a glass-like state. Cryopreservation requires pretreatment, dehydration, cryoprotectant exposure, cooling, storage, and recovery. These processes must be balanced to preserve cryopreserved materials [7,275,276]. Many germplasm preservation scenarios are addressed using different methods. Stored cultures are kept at lower temperatures and light levels to slow metabolism. Gel beads or artificial seeds protect explants from drying or damage during encapsulation. As mentioned, cryopreservation allows long-term storage. Tissue culture and field gene banks preserve plant diversity in vitro and in nature [6,277].

Innovation advances these methods while following established procedures ensures confidence in comparisons. Automation, bioreactors, and gene editing revolutionize horticulture [69,278]. Protocols with molecular markers and omics data optimize culture conditions and genetic stability. New horticulture uses complex methods and strategies. They underpin plant propagation, conservation, and biotechnology advances. Researchers and practitioners are about to revolutionize horticulture by combining established protocols with novel approaches to improve plant production, protection, and genetic enhancement, as shown and described in Figure 2 and Table 6 [279–281].

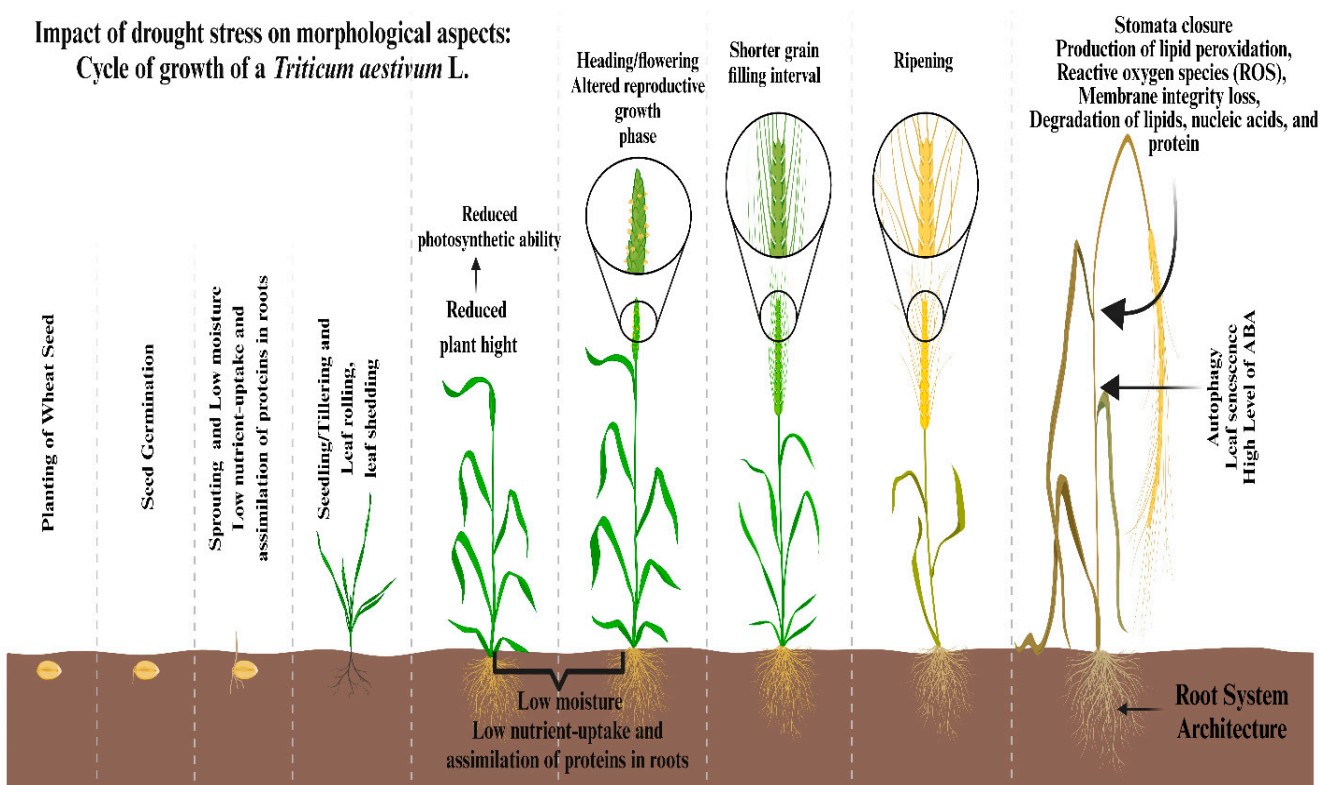

**Figure 2.** Nanoparticles improved stress tolerance in wheat crops from germination to the harvesting stage.

**Table 6.** Effect of Nanoparticle-Mediated Drought Stress Alleviation in Food Crops and Economic Impact.

| Food Crop | Drought Stress Effect without-NPs | Drought Stress Effect with Phyto-Synthesized-NPs | Physiological Impact | Economic Impact | References |
|---|---|---|---|---|---|
| *Triticum aestivum* | Yield decrease (altered metabolism | Enhanced growth improved metabolism | Improved root development, leaf structure | Increased market value, reduced losses | [106,282] |
| *Oryza sativa* | Grain size reduction (nutrient deficiency | Increased grain size improved nutrient content | Improved nutrient uptake, enhanced growth | Cost-effective production, higher profits | [103,283] |
| *Zea mays* | Stunted growth yield reduction | Enhanced growth rate increased yield | Increased photosynthetic efficiency | Enhanced export potential | [284,285] |
| *Glycine max* | Delayed flowering reduced protein content | Timely flowering, improved protein content | Improved seed quality, faster maturation | Improved price competitiveness | [286,287] |
| *Solanum lycopersicum* | Reduced fruit size, loss of flavor | Improved fruit size enhanced flavor | Enhanced taste, nutritional value | Reduced spoilage, better market reception | [288–290] |
| *Hordeum vulgare* | Yield reduction (20%), decreased enzyme activity | Yield increase (30%), normalized enzyme activity | Increased disease resistance | Better yield, economic efficiency | [291–293] |
| *Lactuca sativa* | Slow growth reduced vitamin content | Accelerated growth enhanced vitamin content | Improved texture, taste | Increased consumer acceptance | [294–296] |

**Table 6.** *Cont.*

| Food Crop | Drought Stress Effect without-NPs | Drought Stress Effect with Phyto-Synthesized-NPs | Physiological Impact | Economic Impact | References |
| --- | --- | --- | --- | --- | --- |
| *Solanum tuberosum* | Reduced tuber size delayed harvest time | Increased tuber size on-time harvest | Enhanced tuber quality, uniformity | Higher market value, reduced wastage | [297–299] |
| *Helianthus annuus* | Reduced seed yield decreased oil content | Increased seed yield enhanced oil content | The improved oil quality, seed vigor | Improved production economics | [300–303] |
| *Arachis hypogaea* | Reduced pod size, lower protein concentration | Improved pod size increased protein concentration | Improved seed germination, growth | Enhanced processing efficiency | [302,304–306] |
| *Avena sativa* | Reduced grain yield decreased nutritional value | Increased grain yield enhanced nutritional value | Enhanced resistance to drought, pest | Reduced production costs, increased demand | [307,308] |
| *Prunus dulcis* | Reduced nut size, lower oil quality | Improved nut size increased oil quality | Improved tree health, nut quality | Improved export potential, price stability | [309–311] |
| *Malus domestica* | Reduced fruit quality delayed ripening | Improved fruit quality on-time ripening | Enhanced taste, appearance | Increased market acceptance, demand | [312,313] |

*4.1. Enhancement of Seed Germination and Seedling Growth*

Phyto-synthesized NPs are specifically engineered to regulate seed germination and enhance seed growth during dry weather by facilitating water uptake and modulating osmotic regulations [314]. For example, silver nanoparticles synthesized from plant extracts are pivotal in promoting seed germination and crop growth [315]. The underlying mechanism of these NPs involves nutrient absorption from the soil and subsequent translocation to the plant's stems, branches, and leaves, even under drought stress. This process aids in osmotic adjustments as well [316]. Despite limited water resources, the nanoparticles expedite water absorption and retention within plant tissues, supporting seed germination and early seedling development.

Furthermore, these nanoparticles contribute to maintaining essential turgor pressure and optimizing cellular functions, which are critical for the survival and progression of plants facing drought-induced conditions. A notable example is the utilization of silver nanoparticles (Ag-NPs) derived from plant extracts. Research confirms that these silver NPs significantly enhance seed germination rates and stimulate seedling growth across various cultivated species when subjected to drought stress [148,317,318]. This observation implies that Ag-NPs hold the potential to serve as a promising tool for enhancing plantlet establishment and initial growth, particularly in regions susceptible to water scarcity or erratic precipitation patterns. While this field of inquiry is still relatively nascent, it is essential to acknowledge the need for additional studies to comprehensively elucidate the fundamental mechanisms and potential environmental consequences associated with using phyto-synthesized NPs to bolster plant growth under drought-induced stress [214,319,320]. Nonetheless, the results are promising and underscore the potential for sustained agronomic improvements that could augment yield productivity in challenging environmental conditions. Continued research efforts and meticulous assessments will be indispensable in fully harnessing the benefits of these innovative NPs for agronomic processes while mitigating potential adverse effects on the environment and human health [321,322].

### 4.2. Improvement of Water Relations

In recent research, there has been a growing interest in exploring the potential impacts of using phyto-synthesized NPs as an innovative and biodegradable approach to understanding various aspects of plant physiology. This approach becomes particularly relevant when addressing ecological challenges such as water scarcity [323,324]. One particularly dynamic aspect that has garnered considerable attention is the influence of phyto-synthesized NPs on water-related processes within plants. Through strategic utilization of phyto-synthesized NPs, researchers have uncovered compelling evidence of their positive effects on vital water-related pathways [325]. These effects include enhanced water utilization efficiency and the maintenance of turgor pressure within plant cells. $Fe_3O_4$-NPs have demonstrated favorable outcomes for strengthening drought-induced water retention in plants. These $Fe_3O_4$-NPs can augment hydraulic root conductivity, effectively facilitating the efficient uptake and movement of water throughout the plant structure [251,326].

Consequently, plants treated with $Fe_3O_4$-NPs have improved water utilization efficiency, enabling them to manage available water resources effectively, even under drought-stress conditions [202,327]. The implications of these recent findings are both extensive and profound, particularly in the realm of agriculture. Effective water retention can significantly impact crop yield and overall productivity. By harnessing the potential of phyto-synthesized NPs, farmers and agronomists in water-scarce regions could adopt a sustainable and ecologically sound strategy to enhance crop resilience and mitigate the adverse effects of water deficiency. It has the potential to completely transform farming methods, making them resilient even in arid environments [119,328].

However, despite the remarkable outcomes observed, it is essential to acknowledge that the application of phyto-synthesized NPs in plant systems is still developing. Many aspects require further investigation. A comprehensive understanding of the fundamental mechanisms underlying the interaction between nanoparticles and water is imperative. Furthermore, long-term studies must assess the potential biological effects and unintended consequences of widespread nanoparticle use in agricultural settings [329–331]. Nevertheless, these endeavors collectively indicate that phyto-synthesized NPs possess tremendous potential as a cutting-edge and practical approach to addressing plant water stress. As researchers delve deeper into this captivating field of study, it is anticipated that integrating nanotechnology into crop cultivation will contribute to sustainable and robust crop production methodologies, ultimately advancing global initiatives to ensure food security and ecological sustainability. Phyto-synthesized-NPs promise to ameliorate plant–water relations during drought stress by enhancing water utilization efficiency and maintaining turgor pressure/For instance, $Fe_3O_4$-NPs have been demonstrated to elevate root hydraulic conductivity and improve water use efficiency in plants subjected to drought stress [115,294,297–299].

### 4.3. Stimulation of Photosynthesis and Chlorophyll Content

The stimulation of chlorophyll and photosynthesis within plant cells using phyto-synthesized NPs has garnered significant attention in recent years due to its potential implications for environmental and agricultural sustainability. Photosynthesis is the fundamental process by which plants convert light energy into chemical energy, crucial for their growth and development. At the same time, carbon fixation plays a pivotal role in global ecosystems [119,316,328]. During environmental challenges like drought, plants often encounter reduced photosynthetic rates and chlorophyll degradation, leading to diminished growth and productivity. However, research indicates that applying phyto-synthesized NPs can ameliorate these detrimental effects and enhance photosynthetic efficiency [1,2,185,332–334].

Moreover, phyto-synthesized NPs possess inherent antioxidant properties, which aid in combating the harmful impacts of reactive oxygen species (ROS) that accumulate during drought conditions. ROS can induce oxidative harm to cellular components, including photosynthetic pigments and chloroplasts. By neutralizing the effects of these harmful

radicals, NPs can safeguard chlorophyll molecules from degradation and sustain optimal chlorophyll content within plant leaves [316,328,335]. Phyto-synthesized NPs have been found to modulate gene expression associated with photosynthesis, resulting in an upregulation of essential proteins and enzymes responsible for enhancing the photosynthetic rate. This molecular regulation improves photosynthetic efficiency even under adverse environmental conditions [336,337].

Phyto-synthesized NPs, such as ZnO-NPs, affect photosynthesis and chlorophyll levels differently. The outcomes may depend on nanoparticle concentration, size, and the particular plant species under investigation. As a result, further research is essential to elucidate the mechanisms of action of different NPs and their specific interactions with diverse plant systems, thereby enhancing their applications for promoting photosynthesis [156,224,338]. The potential of phyto-synthesized NPs to modulate photosynthesis and improve chlorophyll content holds significant promise for ecologically sustainable agriculture and environmental remediation. By boosting plant growth and stress tolerance, these nanoparticles could mitigate the adverse impacts of water scarcity, climate change, and other ecological challenges on crop yields and environmental well-being. Despite the promising research in the nanoparticle field, comprehensive studies are required to assess the long-term effects, safety considerations, and practical applications of these nanoparticles in real-world agricultural and ecological contexts [202]. Phyto-synthesized NPs can enhance plants' photosynthesis and chlorophyll content to enhance drought tolerance by optimizing the photosynthetic process and safeguarding chloroplasts against oxidative damage. For instance, ZnO-NPs have been shown to elevate chlorophyll content and improve photosynthetic efficiency in drought-stressed plants [165,320,328].

### 4.4. Induction of Antioxidant Defense Systems

Researchers have been increasingly interested in exploring the potential of phyto-synthesized NPs to mitigate the adverse effects of drought stress on plants. Recent studies have highlighted the beneficial effects of these NPs in inducing antioxidant defense systems in plants [339]. Among the different NPs studied, Au-NPs and $TiO_2$-NPs have shown promising results in enhancing the activities of essential antioxidant enzymes, including superoxide dismutase (SOD), catalase (CAT), and ascorbate peroxidase (APX), when plants are exposed to drought stress [340–342]. The induction of antioxidant defense systems by phyto-synthesized NPs is a significant development in plant stress tolerance research. These NPs help plants counteract the harmful effects of reactive oxygen species (ROS) generated during drought stress, thereby reducing overall oxidative stress levels [343,344].

Consequently, plants exhibit improved growth, development, and survival even under water-limited supply. The exact mechanisms behind the NPs-induced activation of antioxidant defense systems require further investigation. Nevertheless, the findings indicate promising prospects for developing sustainable strategies to enhance plant drought tolerance [209]. Further research and exploration in this field will likely provide novel insights and practical applications for sustainable crop production and environmental conservation [256,345].

### 4.5. Regulation of Plant Hormones

Recently, interest in controlling plant hormones with phyto-synthesized NPs to improve plants' capacity to withstand drought stress has been increasing. Researchers have determined the critical functions of various plant hormones in the responses of plants to stress, including abscisic acid (ABA), salicylic acid (SA), and jasmonic acid (JA). The amounts of these hormones can be influenced by phyto-synthesized NPs when plants are stressed by drought, potentially causing the plants to respond [256,345,346]. Due to their numerous applications in industries, including agriculture and environmental cleanup, phyto-synthesized NPs have emerged as a promising study field, which also plays a significant role in plant hormonal regulations [3,347,348].

Recently, interest has increased in employing phyto-synthesized NPs to regulate plant hormones and improve plants' resistance to drought stress. These nanoparticles are made from plant extracts, which serve as sustainable and eco-friendly reducing and stabilizing agents, providing a more environmentally friendly option to conventional chemical processes [67,349]. The passage also emphasizes the crucial functions of abscisic acid (ABA), salicylic acid (SA), and jasmonic acid (JA) as essential participants in how plants react to stress. These phyto-synthesized NPs can remarkably regulate the levels of these hormones in response to drought stress, which may lead to the onset of adaptive responses in plants. Abscisic acid (ABA) has a significant and multifaceted function in how plants respond to stress, particularly in drought-like conditions. Controlling stomata closure, which aids plants in water conservation by lowering transpiration, is one of its primary roles [3,341,345–348]. When plants are under drought stress, phyto-synthesized NPs can help them conserve water by boosting the levels of ABA. As a signaling molecule, ABA also affects the expression of several genes involved in the stress response. This complex gene regulation system improves the plant's capacity to withstand and respond to adverse environmental conditions, ultimately fostering drought resistance [28]. Although salicylic acid (SA) is widely established to protect plants from biotic stresses, it has recently attracted interest for its ability to help plants in abiotic and drought stresses [2–4,346].

Previous research shows that salicylic acid (SA) protects plants against drought-related damage. It can be affected by phyto-synthesized NPs. For plants to appropriately respond to water constraints, a sophisticated signaling network formed by SA and other stress-related hormones is necessary [5,350]. Jasmonic acid (JA), particularly in drought adaptation, is a crucial aspect of the plant's toolset for managing stress. A series of reactions brought on by regulating JA levels by phyto-synthesized NPs activate genes and signaling pathways linked to drought responses. As a result, the plant can better manage water scarcity and endure adverse environmental circumstances. A fascinating area of study is the relationship between phyto-synthesized NPs and JA signaling, which has the potential to lead to the development of brand-new, ground-breaking techniques for improving drought tolerance in plants and advancing environmentally friendly farming methods, as shown in Figure 3 [324,351].

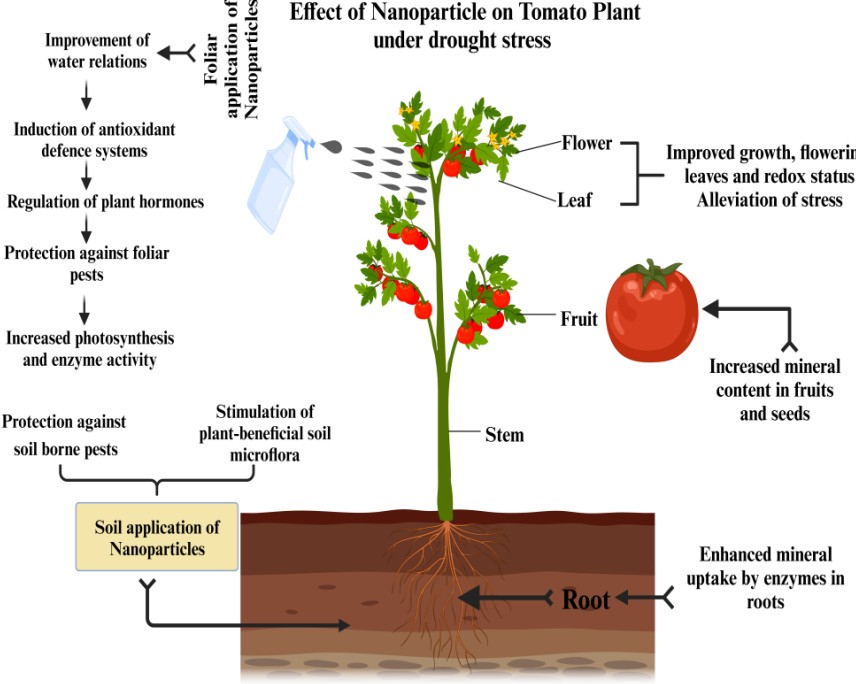

**Figure 3.** Drought-tolerant Tomato crops can be made via nanoparticle-mediated mechanisms and plant hormonal regulations.

## 5. Plant Proteomics and Gene Expression Regulation of Drought Response by Nanoparticles

Nanoparticles like metal nanoparticles (like Silver Nanoparticles), silicon nanoparticles, and carbon nanoparticles (like Carbon Nanotubes) significantly affect how plants react to drought stress at the molecular level. The complicated chemical processes can be summed up by saying that silver nanoparticles can cause oxidative stress by making ROS. On a molecular level, they may raise the expression of genes that code for antioxidant enzymes like SOD and CAT. This makes these protective proteins more active. Silicon nanoparticles interact with guard cells in the stomata. At the level of molecules, this exchange can change how ion channels work and signals are sent. This can lead to better control of the stomata and less water loss. At the molecular level, carbon nanoparticles, especially carbon nanotubes, can stimulate the production of genes that respond to stress. They may increase the activity of transcription factors like DREB and MYB, which then turn on genes that help the cell deal with stress.

The effect of nanoparticles on plants' proteome and gene expression during drought stress is an important research topic. Intricate molecular modifications, such as those affecting nano clays, nano iron oxide, and nano-silica, are essential to understanding these effects. Nano clays, for instance, have been shown to alter proteomes in plants. The overexpression of stress-related proteins such as LEA and chaperones may be involved at the molecular level. Protein modifications like this aid plants in resisting the cellular stress brought on by drought. Nano-sized iron oxide particles change the way soil absorbs nutrients. This may entail alterations in the expression of genes involved in nutrition transport and assimilation at the molecular level. This guarantees that plants can obtain the nutrients they need despite poor conditions. Potential epigenetic alterations induced by silica nanoparticles in plants are being studied. Molecular modifications, such as those to DNA methylation and histone acetylation patterns, can have long-term effects on gene expression. Incorporating these case studies of nanoparticles into the headers provides a more nuanced understanding of the molecular-level interactions between nanoparticles and plants. Stress from drought has caused this. This has ramifications for various physiological processes and adaptive behaviors in plants.

## 6. Potential Risks and Challenges Associated with the Use of Phyto-Synthesized NPs in Agriculture

Using various phyto-synthesized NPs in agriculture provides exciting opportunities to increase crop yield and advance sustainable farming. However, the promise also raises significant issues that we must deliberately address. Concerns about their possible environmental impact and potential harm to organisms that are not their targets are among the main ones [119,352]. Conducting in-depth studies and implementing efficient laws to ensure safe and responsible use is critical. Investigating the long-term impacts of these nanoparticles on soil health and the emergence of resistance in agricultural systems is also necessary [60,353,354].

Moreover, we must consider how small-scale farmers can use and finance this technology and the importance of public perception and education for its widespread acceptance. Phyto-synthesized NPs can be beneficial in developing sustainable agriculture by proactively resolving these issues and encouraging ethical behaviors. To maximize the potential of this novel farming strategy, a balance must be struck between utilizing their advantages and reducing potential risks and challenges, while employing mitigation strategies of using phyto-synthesized NPs in agriculture (Table 7) [119,355–357].

**Table 7.** Risks, Challenges, and Mitigation Strategies of Using Phyto-Synthesized Nanoparticles in Agriculture.

| Aspect | Risks and Challenges | Mitigation Strategies | References |
|---|---|---|---|
| **Food Crops** | 1. Toxicity: Possible toxicity to plants and soil organisms. | Monitoring nanoparticle concentration. | [358,359] |
| | 2. Bioaccumulation: Risk of nanoparticles accumulating in edible parts of plants. | Researching appropriate materials and sizes. | [360,361] |
| | 3. Environmental Impact: Uncontrolled dispersion in the environment. | Implementing controlled release mechanisms. | [362,363] |
| **Drought Management** | 1. Efficiency: Uncertain efficiency in drought resistance. | They are conducting thorough field tests. | [364,365] |
| | 2. Long-term Effects: Unknown long-term impacts on soil health. | Continuous monitoring and adapting practices. | [366,367] |
| | 3. Regulatory Compliance: Legal and regulatory considerations. | They are ensuring compliance with local regulations. | [368,369] |

*6.1. Toxicisty to Non-Target Organisms*

As a popular choice for agricultural usage, phyto-synthesized NPs have attracted considerable attention for their promising capacity to increase plant growth and stress tolerance. As with any new technology, properly assessing dangers posed to agroecosystem non-target creatures is crucial [370]. Responsible use of these phyto-synthesized NPs in agricultural techniques requires weighing their potential advantages against the need to protect the ecosystem. Because of their tiny size and unique features, nanoparticles can interact with living systems differently than larger particles or bulk materials, which raises concerns [371]. When used in farming environments, phyto-synthesized NPs may be ingested by plants and then transferred through the food chain to other creatures, including unintended ones. This prospect necessitates careful investigation and assessment of the potential effects on the ecology of employing these nanoparticles in agriculture [372]. For this technology to be used responsibly and sustainably, it is imperative to understand how these nanoparticles function in living systems. These nanoparticles can have a variety of consequences on species that are not their intended targets, depending on their composition, concentration, and duration of exposure. The overall productivity and sustainability of the agroecosystem might be disrupted, for instance, if soil microorganisms that are crucial for nutrient cycling and soil health are adversely impacted [373].

Similarly, beneficial insects that act as pollinators or natural predators of pests may suffer direct or indirect effects from coming into contact with these nanoparticles. To ensure responsible use in agriculture and safeguard the delicate ecosystem balance, we evaluate these potential effects. Further investigation is required to fully assess the possible toxicity of phyto-synthesized NPs to non-target organisms to ensure their safe and responsible use in agriculture. Before adopting large-scale applications, it is crucial to conduct risk assessments, carefully weighing both the short- and long-term repercussions [114,374–377]. Measures should be taken to reduce any dangers to achieve a balance between using phyto-synthesized NPs advantages for plant development and stress tolerance and safeguarding the agroecosystem's fragile ecological balance. To achieve this equilibrium, eco-friendly behaviors must be emphasized. By adopting a cautious and informed approach, we can protect the environment and non-target creatures while ensuring that using various phyto-synthesized nanoparticles in agriculture is both efficient and sustainable [374,375,378].

*6.2. Impact of Phyto-Synthesized Nanoparticles on Water Tables and Groundwater Quality in Agriculture*

The widespread use of Phyto-synthesized nanoparticles (NPs) in agricultural activities may significantly affect water tables and groundwater quality. A study on their use in

agriculture highlights the possible impact of phyto-synthesized NPs on groundwater levels. This has significant consequences for the future of agricultural water supplies [323]. Long-term exposure of agricultural areas to these nanoparticles raises serious concerns because of the potential for their penetration into the soil and subsequent leaching into groundwater. Subterranean particle movement and behavior may be affected by the introduction of phyto-synthesized NPs, which may change the soil's physicochemical features [374]. This phenomenon's possible influence on groundwater quality is the most pressing concern. Water chemistry may be altered due to interactions between these nanoparticles and minerals and organic matter in the soil as they move through the soil profile [379]. Groundwater quality in these agricultural regions may suffer significantly if these changes are allowed to occur. A further concern is that nanoparticles in groundwater may upset the equilibrium of microbial communities in aquifers. In groundwater ecosystems, microorganisms are crucial to nutrient cycling and water filtration [380]. These vital ecosystem services may be compromised by the persistence of NPs in these habitats, reducing microbial diversity and abundance. A thorough evaluation of the possible impact on water tables and groundwater quality is necessary since phyto-synthesized NPs show promise in many agricultural applications, such as crop protection and nitrogen control [381]. Careful risk assessment and management are essential for the benefits of NP used to outweigh the hazards they cause to water supplies and the environment [382]. A careful equilibrium between utilizing the benefits of phyto-synthesized NPs and protecting the stability of water tables and groundwater quality in agricultural regions is required to promote responsible and sustainable farming activities [383–385].

*6.3. Environmental Fate and Transport*

Due to their rising utilization and potential ecological effects, there has been a significant interest in researching the environmental fate and transportation of phyto-synthesized NPs in recent years. Understanding how phyto-synthesized NPs act in the environment becomes crucial to judging potential hazards [386]. According to previous studies, phyto-synthesized NPs can penetrate the soil profile and even reach subsurface water reservoirs. Further worries regarding water pollution are raised because they have access to numerous water bodies. For informed application decisions and to ensure the protection of our natural resources, it is crucial to comprehend how these nanoparticles behave in the environment. Numerous factors affect phyto-synthesized NPs' rate and transport, such as their stability, surface charge, and physicochemical properties [387].

Moreover, ecological conditions, including pH levels, soil type, and moisture content, play an essential role in their performance. Though nanoparticles move through diverse environmental situations, they can intermingle with many other organisms and are also interlinked with biochemical processes, promoting potential ecosystem modifications [388]. Notwithstanding the critical development in this field, holes linger in our understanding of the long-lasting performance and probable impacts of phyto-synthesized NPs in the environment. Additional widespread studies are required to determine their future, relations, and perseverance on different environmental grounds. This evidence is important for emerging actual danger calculations and managing policies to defend living organisms and the environment from harmful things allied with the practice of phyto-synthesized NPs [103,106,283].

## 7. Future Research Directions and Recommendations

Future research on phyto-synthesized NPs has enormous promise for improving agriculture and tackling issues with global food security. These nanoparticles, made from plant extracts, have the extraordinary potential to encourage plant growth, increase stress tolerance, and improve nutrient intake. We must examine numerous critical research paths and suggestions before profiting from these phyto-synthesized NPs in agriculture [185,332,386].

### 7.1. Optimizing the Synthesis and Application of Phyto-Synthesized NPs

The development of phyto-synthesized NP synthesis is a significant topic that needs more study. Although a method for producing these nanoparticles currently exists, scientists should concentrate on developing more effective, stable, and environmentally acceptable agricultural versions [389]. To do this, we must investigate fresh ideas and strategies for optimizing the synthesis procedure to produce higher yields and higher-caliber nanoparticles. It would also be very beneficial to research how various elements, such as plant species, extract concentrations, and reaction circumstances, affect the synthesis of NPs. We may streamline the procedure and ensure we consistently and reliably create these nanoparticles for agricultural use consistently and reliably by comprehending these elements [83]. Understanding how phyto-synthesized NPs promote plant growth and stress tolerance is another crucial topic that requires research. Our main goal should be to study the interactions between these nanoparticles and plant cells and how they affect different physiological and biochemical pathways. This understanding allows us to create nanoparticles suited for various applications [16]. Scientists can use cutting-edge approaches, including genomics, proteomics, and metabolomics, to explore the complex molecular processes of boosting plants with NPs. It is critical to evaluate the potential environmental effects of phyto-synthesized NPs because their use in agriculture is growing. As we investigate more extensive services for these nanoparticles in farming and other processes, we must ensure they are secure and sustainable [390].

Further research is required to evaluate the effects of these nanoparticles on soil health, water purity, and other ecosystem creatures that are not the primary targets. Scientists may ensure that using phyto-synthesized NPs is consistent with sustainable agriculture methods and prevents unintentional ecological disruptions by understanding the broader ecological ramifications. It is also critical to address the issue of scaling up nanoparticle production [391]. A more profound knowledge of these nanoparticles' benefits and limitations can be attained by assessing how well they perform regarding crop output, nutrient uptake, and stress resistance in various environmental settings and farming methods. Farmers and politicians who want to include phyto-synthesized NPs in their agricultural practices might benefit significantly from the findings of these comparative studies as excellent recommendations [80,330].

### 7.2. Elucidating the Mechanisms of Action of Phyto-Synthesized NPs

In recent years, phyto-synthesized NPs have gained recognition as a promising approach for transforming agriculture by enhancing plant development and stress tolerance. The sustainable green synthesis of NPs using plant molecules efficiently replaces conventional chemical processes. Phyto-synthesized NPs hold much promise, but it is essential to comprehend the intricate mechanisms underlying their beneficial effects on plants. According to recent studies, creating nanoparticles from metal ions involves using various plant parts, including leaves, roots, stems, and flowers [358,360,361]. These plant extracts contain a variety of phytochemicals, including flavonoids, polyphenols, alkaloids, and proteins, which function as reducing and stabilizing agents to create NPs with distinctive physicochemical properties. These NPs may easily be absorbed by plant tissues after being generated, either through root absorption or foliar application, and then transported to various plant organs where they exert their positive effects. According to studies, the phyto-synthesized (NP) size, shape, and surface charge are crucial factors in cellular internalization and the reactions that follow in plant cells [156,338,343,344].

Plant growth and development are facilitated by the modulation of various physiological and biochemical processes at the cellular level by phyto-synthesized NPs. Phyto-synthesized NPs, for instance, can boost root and shoot length, accumulate biomass, and encourage overall plant growth, among other things. The presence of phyto-synthesized NPs in the rhizosphere also affects root architecture by encouraging the development of lateral roots and the proliferation of root hairs, improving nutrient and water uptake efficiency. Additionally, phyto-synthesized NPs have been shown to modify the activ-

ity of enzymes involved in antioxidant defense mechanisms, reducing the cellular damage brought on by oxidative stress. Plants treated with NPs, therefore, show increased resistance to abiotic stressors like drought, salt, heavy metal toxicity, and temperature fluctuations [119,202,326,327].

Additionally, plants exposed to NPs have demonstrated improved resistance to various biotic stressors, including infections caused by bacteria, fungi, and viruses. Understanding these cellular reactions to optimize NP applications in agriculture and customize them to particular crop and stress settings is essential. Furthermore, the signaling pathways in plants activated by phyto-synthesized NPs are also interesting [309,311].

*7.3. Investigating Long-Term Effects and Potential Risks*

Long-term research is essential to fully evaluate the possible dangers related to the use of phyto-synthesized NPs in agriculture. While this preliminary study suggests that phyto-synthesized NPs have positive benefits, further research is necessary to identify any unforeseen effects that might develop in the future. These in-depth investigations ought to concentrate on assessing how phyto-synthesized NPs affect the health of the soil, unintended organisms, and the functioning of the ecosystem as a whole. It is crucial to comprehend how phyto-synthesized NPs affect the condition of the soil. As the foundation of agricultural systems, soils support plant development and the cycling of nutrients [6,322,332]. Although preliminary findings may point to beneficial effects on soil properties, such as increased nutrient availability or improved water retention, prolonged exposure may change the soil's microbial community and nutrient dynamics, impacting crop productivity and soil fertility [25]. Ag-NP treatments had positive short-term effects on some soil parameters. Still, long-term exposure led to decreased soil microbial diversity and altered enzyme activity, raising concerns about the long-term health and fertility of the soil [85]. Long-term studies must include an analysis of the impact of phyto-synthesized NPs on non-target organisms in the agroecosystem. These species—beneficial insects, pollinators, earthworms, and other fauna—are crucial for maintaining ecological balance [77]. The possible influence of phyto-synthesized NPs on the general health of ecosystems must also be considered in long-term investigations.

Plants, animals, microbes, and abiotic elements interact in intricate networks in agricultural systems. Introducing new materials, such as NPs, into these systems may upset the delicate balance of ecosystem dynamics [316]. Copper oxide nanoparticles (CuO-NPs) made from plant extracts were the subject of a ten-year study by Qamer et al. (2021) that examined how they affected an agricultural ecosystem. The study found that while CuO-NPs initially increased agrarian yields, prolonged exposure caused changes in soil microbial communities, decreased plant variety, and impacted insect populations, which reduced the resilience of the ecosystem as a whole [119,330,392]. Long-term research should consider and propose mitigation measures to mitigate negative impacts and assure sustainable usage of phyto-synthesized NPs in agriculture in light of these potential dangers. Based on the study's findings, specific tactics might be suggested, including implementing buffer zones, optimizing NP application rates, and choosing plant species with a low environmental impact. Phyto-synthesized NPs can be used responsibly and intelligently to the fullest extent possible to develop resilient and environmentally friendly agricultural systems [330].

*7.4. Developing Guidelines and Regulations for the Safe Use of Phyto-Synthesized NPs in Agriculture*

Due to its potential to drastically improve crop productivity and pest control, phyto-synthesized NPs have attracted much interest in the agricultural sector. These phyto-synthesized NPs are environmentally friendly substitutes for typical chemical inputs because they utilize environmentally friendly processes incorporating plant extracts [80]. Despite their apparent benefits, their extensive usage raises questions about potential environmental and public health hazards, demanding detailed policies and regulations for sustainable and safe use. One of the leading ecological hazards is the possible buildup of

phyto-synthesized NPs in the soil, which may impact soil health and microbial communities [393]. Another concern is the runoff of NPs into water bodies, which could cause aquatic toxicity and disturb marine ecosystems.

Additionally, a careful examination of phyto-synthesized NPs' effects on non-target organisms, such as beneficial insects and birds, is necessary to guarantee their safety. Occupational exposure for farmers and agricultural employees who handle NPs during application is the primary source of worry regarding human health. In addition, concerns about food safety are raised by the possibility that NPs could be absorbed by crops and then enter the food chain [394]. Furthermore, one factor that must be carefully considered to protect public health is indirect exposure to consumers who consume treated crops or food items. It is crucial to design efficient solutions to reduce these hazards. One strategy specifies safe dosages and application rates for phyto-synthesized NPs. Maximum levels of NPs that can be used in agriculture must be set so that crops, the earth, and the environment do not take damage in ways that were not expected [320,328,332].

Additionally, adding phyto-synthesized NPs to current Integrated Pest Management (IPM) techniques can lower overall chemical inputs and improve those chemicals and responsible applications. The monitoring protocol implementation is essential for the ongoing evaluation of phyto-synthesized NP dispersion and potential environmental effects. Regular soil, water, and air quality monitoring are required to track NP distribution and fate in agricultural environments. Important information on the presence and potential effects of phyto-synthesized NPs will also be provided by biomonitoring studies that examine the accumulation of NPs in plants and other non-target organisms [15,395]. Rigorous risk assessment frameworks must be used in addition to monitoring to assess the possible threats to the environment and public health posed by phyto-synthesized NPs in agriculture. Standardized methods for phyto-synthesized NP exposure evaluation can aid in evaluating the effects of NPs on ecosystems.

In contrast, in-depth assessments of the possible health concerns of NP exposure for consumers, agricultural workers, and farmers should be conducted. Phyto-synthesized NPs show great promise as environmentally beneficial agrarian solutions. However, their responsible adoption necessitates the creation of thorough rules and regulations [13,14,396]. The agricultural sector can take advantage of the advantages of nanotechnology while preserving the environment and public health by addressing potential environmental and human health risks by establishing appropriate application rates, monitoring protocols, and conducting thorough risk assessments [12,397].

## 8. Conclusions

Encouraging outcomes for the use of phyto-synthesized NPs in agriculture have been shown to reduce the negative impacts of drought stress on food crops. The importance of green synthesis techniques, which use plant extracts to create nanoparticles, has been highlighted in this review as an economical and environmentally responsible technique. The many uses of phyto-produced NPs, including Ag-NPs, Au-NPs, $Fe_3O_4$ NPs, Cu-NPs, ZnO-NPs, and $TiO_2$-NPs, have shown their potential to improve plant growth and biochemical characteristics under drought-stress conditions. According to the results of numerous research studies, phyto-synthesized NPs can significantly enhance seed germination and seedling growth, control water relations, promote photosynthesis, increase chlorophyll content, and activate antioxidant defense mechanisms in plants under drought stress. Additionally, these nanoparticles' control of plant hormones offers a valuable defense against the effects of drought stress and raises agricultural output. However, it is essential to consider the dangers and difficulties that could arise from using phyto-synthesized NPs in agriculture. The toxicity of these nanoparticles to non-target organisms and their possible effects on soil health must be carefully considered. To avoid any adverse impact on ecosystems, it is crucial to comprehend the environmental destiny and transit of these phyto-synthesized NPs. Future research should concentrate on refining synthesis

processes to boost nanoparticle stability and bioavailability to ensure the safe and efficient deployment of phyto-synthesized NPs in agriculture.

Furthermore, it is critical to obtain a more precise knowledge of the underlying mechanisms through which these phyto-synthesized NPs affect plants. Long-term research is required to evaluate the persistence and potential cumulative impacts of phyto-synthesized NPs in agroecosystems. Creating detailed policies and regulations for using phyto-synthesized NPs ethically in agriculture is crucial. By doing this, it will be possible to optimize positive effects while limiting negative ones on the environment, non-target creatures, and human health. Especially in drought-stressed settings, incorporating phyto-synthesized NPs in agricultural techniques holds considerable promise for sustainable crop production. To fully realize the potential of these cutting-edge nanomaterials in agriculture for a robust and secure future of food, however, more research, risk analysis, and policy implementation are required.

**Author Contributions:** Conceptualization, A.W. and M.M.; software, A.W., R.M.M. and W.Z.; validation, A.W., F.B. and W.Z.; resources, M.N.; writing—original draft preparation, A.W., M.M., F.B. and W.Z.; writing—review and editing, A.W., M.M. and W.Z.; visualization, A.W., M.N. and M.M; supervision, W.Z. and A.W. All authors have read and agreed to the published version of the manuscript.

**Funding:** This research received no external funding.

**Institutional Review Board Statement:** Not applicable.

**Informed Consent Statement:** Not applicable.

**Data Availability Statement:** Not applicable.

**Conflicts of Interest:** The authors declare no conflict of interest.

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
