# Peer review of "Current Knowledge, Research Progress, and Future Prospects of Phyto-Synthesized Nanoparticles Interactions with Food Crops under Induced Drought Stress"

_sustainability, doi:10.3390/su152014792_

Round 1

Reviewer 1 Report

The Manuscript ID Sustainability - 2618903 “Current Knowledge, Research Progress, and Future Prospects of Phyto-Synthesized Nanoparticles Interactions with Food Crops under Induced Drought Stressdescribes an interesting review. However, there are several issues that the authors need to address before this manuscript can be published. Here are my comments:

Title: Check the use of upper and lower case in some words such as "of", "with" etc.

Line 22: Drought detrimental impact ->Detrimental impact of drought

Line 23: Write nanoparticles in full for the first mention and subsequent ones use abbreviations. For example line 23; phyto-synthesized (NPs) -> phyto-synthesized nanoparticles (NPs).

Lines 25, 31 and many sections throughout the manuscript: remove brackets from (NPs); phyto-synthesized (NPs) -> phyto-synthesized NPs.

The abstract needs to be revised to capture clearly the detrimental impact of drought, why nanoparticles and not any other method for drought tolerance, mechanism of action of nanoparticles, research gaps that needs to be filled in future studies and conclude by indicating why this review is important.  

Lines 44, 45: delete the two sentences. Not important.

Under the introduction part: it is important to have a snapshot of drought stress  and its effects, yield losses as well as the processes disrupted under drought stress conditions.

1.1  Drought stress

1.2  Why phyto-synthesized (NPs) are so promising

1.3  Review scope and approach

Line 94: Section 1.2: Replace “Objective and scope” with “Review scope and approach”

-The review covers published literature between which years - for example between 2010 and 2023 (until March)

            -The search was performed in which databases?

            -What was the review approach?

-Decision-making process for the selection of appropriate journal articles and the scope of the review.

Line 95: delete “exhaustive”

Line 99: global changes -> global changes in climate or global climate change.

Line 102: What do you mean by “neutral?”

Table 1: The table is not self-explanatory. Please elaborate the title of the table so that the table is self-explanatory.

Line 177: Delete “comprehensive” for the table title because the list is not exhaustive.

Table 4: Examples of the food crops are not captured in the table. Please indicate the food crops on which the technology has been demonstrated.

Sub-Sections 3.1 – 3.6: Why was characterization of nanoparticles not captured? This should be captured.

In sub-sections 3.1 – 3.6: there is a repetition of the mechanism of action of the nanoparticles, which is supposed to be captured in section 4. Critically revise to delete all the repetitions on the mechanism of action of each of the different nanoparticles. In sub-sections 3.1 – 3.6, provide information on synthesis and characterization of nanoparticles.

What is the mode of uptake and transport of nanoparticles in plants?

Section 4: The information provided in this section is describing the effects or role of nanoparticles in drought stress alleviation and not mechanisms. Please revise, the information provide is not on mechanisms. The authors can revise the titles of section and also subsections.

Section 4: The effect on nanoparticles on molecular aspects/gene expression is not captured? Why and yet there is a lot of published literature on the same. In addition, the effect of nanoparticles on drought stress alleviation based on proteomics has not been captured.

Lines 731 – 761: subsection 6.2 is a repetition of what has already been covered in section 4. Delete this subsection (6.2), unless the authors provide specific gaps which need to be investigated in future studies.

Minor editing of English language.

Author Response

RESPONSES TO THE COMMENTS (REVIEWER-1)

The comments by the Reviewers are in BLACK, and the responses from the author are in

Blue.

REVIEWER 1 COMMENTS

Author's Responses

Title: Check the use of upper and lower case in some words such as "of," "with," etc.

Our title has been revised with "of" and other class words. We understand your concern and have modified the title to make it more structured and easier to understand.

2. Drought detrimental impact

The detrimental impact of drought stress.

3. Line 23: Write nanoparticles in full for the first mention, and subsequent ones use abbreviations. For example, line 23: phyto-synthesized (NPs) -> phyto-synthesized nanoparticles (NPs).

"Correct as suggested " phyto-synthesized nanoparticles (NPs)

4. Lines 25, 31, and many sections throughout the manuscript: remove brackets from (NPs); phyto-synthesized (NPs) -> phyto-synthesized NPs.

"Correct as suggested " phyto-synthesized NPs

5. The abstract needs to be revised to capture the detrimental impact of drought, why nanoparticles and not any other method for drought tolerance, the mechanism of action of nanoparticles, research gaps that need to be filled in future studies, and the conclusion by indicating why this review is essential.

Done as suggested."

6. Lines 44, 45: delete the two sentences. Not important.

"Done as suggested."

7. Line 94: Section 1.2: Replace “Objective and scope” with “Review scope and approach.”
-The review covers published literature between which years - for example, between 2010 and 2023 (until March)

-The search was performed in which databases?

-What was the review approach?

-Decision-making process for the selection of appropriate journal articles and the scope of the review

 We reviewed the most recent article concerning nanoparticle properties to prepare for this review. We compared it to current and future perspectives. This gap between 2017 and 2023, i.e., Google Scholar, PubMed, Elsevier, NCBI official, etc.,

8. Line 99: global changes -> global changes in climate or global climate change

"Done as suggested."

9. Table 1: The table is not self-explanatory. Please elaborate on the title of the table so that the table is self-explanatory.

Table 1. Comparative Analysis of the Green Synthesis of Nanoparticles: Methodological Approaches and Implications for Sustainability and Practical Applications.
“Revised as per the reviewer's suggestion.”

10. Line 177: Delete “comprehensive” for the table title because the list is not exhaustive.

"Done as suggested."

11. Table 4: Examples of the food crops are not captured. Please indicate the food crops on which the technology has been demonstrated.

"Done as suggested."

Table 4. Types of Nanoparticles and Their Mechanisms of Action for Mitigating Drought Stress in Specific Food Crops.

12. Sub-Sections 3.1 – 3.6: Why was characterization of nanoparticles not captured? This should be charged.

13. In sub-sections 3.1 – 3.6, there is a repetition of the mechanism of action of the nanoparticles, which is supposed to be captured in section 4. Critically revise to delete all the repeats on the mechanism of action of each nanoparticle. Sub-sections 3.1 – 3.6 provide information on the synthesis and characterization of nanoparticles.

"Done as suggested."

14. What is the mode of uptake and transport of nanoparticles in plants?

Done as suggested."

15. The information provided in this section describes nanoparticles' effects or role in drought stress alleviation, not mechanisms. Please revise; the information provided is not on tools. The authors can edit the titles of sections and subsections

"Done as suggested."

176 Section 4: The effect of nanoparticles on molecular aspects/gene expression is not captured? Why and yet there is a lot of published literature on the same. In addition, the effect of nanoparticles on drought stress alleviation based on proteomics has not been captured.

"Done as suggested."

Reviewer 2 Report

Specific comments:

1). The English language of the MS needs minor checks.

2) The title of the review should be repharased to match with the content of the MS.

3) Rewrite the abstract. Authors should follow journal guide and reduce the abstract to 200 words.

4) The statement problem of the review should be clearly stated in the Introduction section.

5) Tables should be formatted by removing the gridlines.

6) Since it is a future prospect and recommendation, why the reference citations and subsections in L693-839?

Minor comments.

L53 and 94: delete

L693 and 840: check the section number ‘6'

The English language needs minor checks.

Author Response

Review Report Form 02 (Red Color)

Comments and Suggestions for Authors

Specific comments:

1). The English language of the MS needs minor checks.

Author response: MS Has been reviewed and addressed all minor mistakes and is well structured according to reviewer suggestions

2) The title of the review should be rephrased to match the content of the MS.

Author response: Tilte has been addressed and well structured according to reviewer suggestions

3) Rewrite the abstract. Authors should follow the journal guide and reduce the abstract to 200 words.

Author response: Follow the journal guide and reduce the abstract; we rewrite the abstract, and the wording limits are reduced.

4) The statement problem of the review should be clearly stated in the Introduction section.

Author response: Done as per suggestions.

5) Tables should be formatted by removing the gridlines.

Author response: Done as per suggestions.

6) Since it is a future prospect and recommendation, why the reference citations and subsections in L693-839?

Author response: To understand and elaborate on its importance, we divided it into subsections to shed light on each section.

Minor comments.

L53 and 94: delete

Author response: Done as per suggestions.

Reviewer 3 Report

Please include from what bibliographic databases you got the bibliography for this review and what strategy you had to find the relevant references.

Lines 59-63 please cite authors

Lines 67-69 please cite authors

Lines 144-166 please cite authors

Lines 85-86 sentence is incomplete

Line 1632 incomplete reference

Author Response

Review Report Form 03 (Green Color)

Open Review

Point 01: Please include from what bibliographic databases you got the bibliography for this review and what strategy you had to find the relevant references.

Author response: We reviewed the most recent article concerning nanoparticle properties to prepare for this review. We compared it to current and future perspectives. This gap between 2017 and 2023, i.e., Google Scholar, Pubmed, Elsevier, NCBI official, etc., must be collectively called the respective article.

Point 02: Lines 59-63, please cite authors.

Author response: We acknowledge your point for information regarding lines 59-63. Your inquiry has been made, and the appropriate citation is cited.

Point 03: Lines 67-69, please cite authors.

Author response: We acknowledge your point for information regarding lines 67-69. Your inquiry has been made, and the appropriate citation is cited.

Point 04: Lines 144-166, please cite authors.

Author response: We acknowledge your point for information regarding lines 144-166. Your inquiry has been made, and the appropriate citation is cited.

Point 05: Lines 85-86 The sentence is incomplete.

Author response: Done as suggested.

Point 06: Line 1632 incomplete reference

Author response: Done as suggested.

Manuscript ID: sustainability-2618903 –

Specific Comments

  1. What does the research address the main question?

 The main question addressed in this research is the current state of knowledge, research Progress, and prospects of Phyto-Synthesized Nanoparticle interactions with food crops under induced drought stress conditions.

  1. Do you consider the topic original or relevant in the field? Does it address a specific gap in the area?

Yes, the topic is original and very relevant because Phyto-Synthesized Nanoparticles have many applications in agriculture with good results in increasing yields and quality. Fortunately, the paper also warns of the risks of applying without knowing the side effects.

  1. What does it add to the subject area compared with other published material?

 It adds a complete review integrating the current state of knowledge, research Progress, and future prospects of Phyto-Synthesized Nanoparticle interactions with food crops under induced drought stress.

  1. What specific improvements should the authors consider regarding the methodology?

What further controls should be considered?

 It has to be considered that a review paper has a distinct methodology compared to a research paper. Still, in the document, I need to include a description of what different bibliographic databases have been used by the authors.

Author response: We reviewed the most recent article concerning nanoparticle properties to prepare for this review. We compared it to current and future perspectives. This gap between 2017 and 2023, i.e., Google Scholar, PubMed, Elsevier, NCBI official, etc.,

  1. Are the conclusions consistent with the evidence and arguments, and do they address the central question?

Yes, they are consistent and also include the important point that more research, risk analysis, and policy implementation will be required for a robust and secure future of food production.

  1. Are the references appropriate?

 Yes, the references are appropriate and include books and book chapters. However, some of the authorities must be adapted to the style of the Sustainability journal.

  1. 7. Please include any additional comments on the tables and figures.

 The figures are well illustrated, and the tables are sufficiently explicative.

Round 2

Reviewer 2 Report

Revised manuscript received.